# Difference in miRNA Expression in Functioning and Silent Corticotroph Pituitary Adenomas Indicates the Role of miRNA in the Regulation of Corticosteroid Receptors

**DOI:** 10.3390/ijms23052867

**Published:** 2022-03-05

**Authors:** Beata J. Mossakowska, Paulina Kober, Natalia Rusetska, Joanna Boresowicz, Maria Maksymowicz, Monika Pękul, Grzegorz Zieliński, Andrzej Styk, Jacek Kunicki, Tomasz Mandat, Mateusz Bujko

**Affiliations:** 1Department of Molecular and Translational Oncology, Maria Sklodowska-Curie National Research Institute of Oncology, 02-781 Warsaw, Poland; beata.mossakowska@pib-nio.pl (B.J.M.); paulina.kober@pib-nio.pl (P.K.); natalia.rusetska@pib-nio.pl (N.R.); joanna.boresowicz@gmail.com (J.B.); 2Department of Cancer Pathomorphology, Maria Sklodowska-Curie National Research Institute of Oncology, 02-781 Warsaw, Polandmonika.pekul@pib-nio.pl (M.P.); 3Department of Neurosurgery, Military Institute of Medicine, 04-141 Warsaw, Poland; gzielinski@wim.mil.pl (G.Z.); astyk@wim.mil.pl (A.S.); 4Department of Neurosurgery, Maria Sklodowska-Curie National Research Institute of Oncology, 02-781 Warsaw, Poland; jkunickii@gmail.com (J.K.); tomasz.mandat@pib-nio.pl (T.M.)

**Keywords:** neuroendocrine pituitary tumors, Cushing’s disease, silent corticotroph adenoma, miRNA, hsa-miR-124-3p, *NR3C1*, glucocorticoid receptor

## Abstract

Corticotroph pituitary adenomas commonly cause Cushing’s disease (CD), but some of them are clinically silent. The reason why they do not cause endocrinological symptoms remains unclear. We used data from small RNA sequencing in adenomas causing CD (*n* = 28) and silent ones (*n* = 20) to explore the role of miRNA in hormone secretion and clinical status of the tumors. By comparing miRNA profiles, we identified 19 miRNAs differentially expressed in clinically functioning and silent corticotroph adenomas. The analysis of their putative target genes indicates a role of miRNAs in regulation of the corticosteroid receptors expression. Adenomas causing CD have higher expression of hsa-miR-124-3p and hsa-miR-135-5p and lower expression of their target genes *NR3C1* and *NR3C2*. The role of hsa-miR-124-3p in the regulation of *NR3C1* was further validated in vitro using AtT-20/D16v-F2 cells. The cells transfected with miR-124-3p mimics showed lower levels of glucocorticoid receptor expression than control cells while the interaction between miR-124-3p and *NR3C1* 3′ UTR was confirmed using luciferase reporter assay. The results indicate a relatively small difference in miRNA expression between clinically functioning and silent corticotroph pituitary adenomas. High expression of hsa-miR-124-3p in adenomas causing CD plays a role in the regulation of glucocorticoid receptor level and probably in reducing the effect of negative feedback mediated by corticosteroids.

## 1. Introduction

Pituitary adenomas (also referred to as pituitary neuroendocrine tumors, PitNETs) represent about 10–20% of intracranial neoplasms in adults. They may originate from different kinds of secretory pituitary cells including corticotroph ACTH-secreting cells. Corticotroph adenomas commonly cause ACTH-dependent Cushing’s disease, but a significant proportion of these tumors are endocrinologically non-functioning and classified as subclinical/silent corticotroph adenomas (SCAs) [1].

CD-causing ACTH tumors are commonly small microadenomas with approximately 50% being smaller than 5 mm, which is challenging for MRI diagnostics [2]. In contrary, SCAs are commonly diagnosed due to neurological symptoms related to tumor mass at the stage of large macroadenomas. Frequently they show invasive growth and increased proliferation index [1]. According to current recommendations, SCAs are now referred to as “high-risk” pituitary adenomas which refers to their fast and invasive growth, high risk of recurrence and resistance to medical therapy [3,4]. They are recognized to be more aggressive than other clinically nonfunctioning pituitary tumors such as those of gonadotroph origin or null-cell adenomas [5].

The mechanism underlying the difference in secretory activity of CD-causing and subclinical tumors is unclear and only a few studies focused on this issue were published. The results indicated a role of the expression levels of particular genes/proteins involved in the regulation of POMC expression and pro-hormone conversion into ACTH as well as genes involved in pituitary differentiation [6,7,8,9,10,11,12,13]. However, it also appears that both active and silent corticotroph adenomas share a similar overall gene expression profile [14,15].

The aim of this study was to compare the profiles of microRNA (miRNA) expression in clinically functioning and silent corticotroph adenomas and to identify miRNAs that play a role in different ACTH secretory activity.

## 2. Results

### 2.1. Patients Characteristics

The study included 28 patients with CD and 20 patients suffering from SCA. All patients with CD had clear clinical signs and symptoms of hypercortisolism verified according to biochemical criteria including elevated midnight cortisol levels and 24 h urinary free cortisol (UFC). Patients with SCA had no clinical or biochemical signs of hypercortisolism and showed normal levels of midnight cortisol and 24 h UFC. Patients with CD had significantly higher morning serum cortisol levels than patients with SCAs (*p* = 0.0002) while no significant difference was observed in the morning serum ACTH levels. No difference in cortisol/ACTH ratio was observed between CD and SCA patients.

All the adenoma samples were ACTH-positive upon immunohistochemical staining against pituitary hormones (ACTH, GH, TSH, FSH, LH, α-subunit) and had characteristic ultrastructural features of corticotroph adenoma. Forty-one adenomas were positive only for ACTH, while seven ACTH-positive adenomas showed additional moderate/weak immunoreactivity for α-subunit. Increased proliferation assessed by Ki67 index ≥ 3% was observed in a similar proportion of CD and SCA patients, seven tumors causing CD and five SCAs. A higher proportion of sparsely vs. densely granulated adenomas was observed in SCAs than in CD-related adenomas, but the difference did not cross a significance threshold (*p* = 0.0787). No difference in the proportion of invasive/noninvasive adenomas was observed in clinically functioning and silent corticotroph adenomas.

All SCAs were macroadenomas, while tumors causing CD included 17 macroadenomas and 11 microadenomas. No significant differences in preoperative clinical parameters, including 24 h UFC, morning serum ACTH level, morning and midnight serum cortisol level, cortisol/ACTH ratio, were observed between CD patients with micro- and macroadenomas. Irrespectively, a correlation between tumors size and ACTH level (Spearman R= 0.4678; *p* = 0.0121) and a negative correlation between cortisol/ACTH ratio (Spearman R= −0.4015; *p* = 0.0342) was observed in CD patients.

No correlation was found between the remaining biochemical parameters and tumor size. Overall, the patients’ characteristics are presented in Table 1, while details including both the clinical and histopathological data are shown in Appendix A.

### 2.2. Identification of miRNAs Differentially Expressed in Corticotroph Adenomas Causing CD and Subclinical Cortiotroph Adenomas

NGS data on miRNA expression of 48 corticotroph adenomas from previous investigation were used to compare miRNA expression levels between adenomas causing CD (*n* = 24) and subclinical corticotroph adenomas (*n* = 20). Sequencing of small RNA libraries produced approximately 2,497,367 reads per sample, which were mapped to the human genome (hg19) and used for quantification of expression levels of known miRNAs, according to miRBase 22 release. Sequencing reads were annotated to 1917 miRNAs. Measurements of 1902 mature miRNAs expression were included in the analysis, after filtering out the miRNAs with low expression.

When miRNA profiles of adenomas causing CD and SCAs were compared, a total of 19 differentially expressed miRNAs were found that met the criteria of adjusted *p*-value < 0.05. This set included 16 miRNAs with higher expression in tumors causing CD: hsa-miR-129-2-3p, hsa-miR-129-5p, hsa-miR-124-3p, hsa-miR-132-5p, hsa-miR-129-1-3p, hsa-miR-135b-5p, hsa-miR-27a-3p, hsa-miR-10b-5p, hsa-miR-9-3p, hsa-miR-6506-3p, hsa-miR-6864-5p, hsa-let-7b-5p, hsa-miR-670-3p, hsa-miR-22-5p, hsa-miR-346 and hsa-miR-9-5p, Three miRNAs with lower expression in CD patients were found: hsa-miR-1909-3p, hsa-miR-4319 and hsa-miR-181b-3p. Details are presented in Table 2 and Figure 1A,B.

### 2.3. The Correlation of miRNA Expression and Patients’ Clinical Data

Since the clustering of the tumors based on the expression of differentially expressed miRNAs did not clearly separate functioning and silent adenomas, we determined whether the expression of the identified differentially expressed miRNAs is directly related to the results of patients’ laboratory tests as well as *POMC* expression, measured in tumor samples with qRT-PCR. For this purpose, Spearman’s correlation was applied to calculate a correlation matrix. We observed a significant positive correlation between 13 miRNAs out of 19 differentially expressed miRNAs and at least one of clinical laboratory parameters: serum ACTH, morning cortisol level, midnight cortisol level or 24 h UFC. For 11 miRNAs, with higher expression in patients with CD a positive correlation was observed, while a negative correlation was observed for 3 miRNAs that have lower expression in patients with CD. Four of the differentially expressed miRNAs, hsa-miR-9-3p, hsa-miR-9-5p, hsa-miR-27a-3p and hsa-miR-6506-3p, are correlated with *POMC* expression level in tumor tissue. The absolute value of correlation coefficient ranged between 0.31 and 0.55 which indicates a weak/moderate relationship. Details are presented in Figure 1C.

### 2.4. Funtional Enrichment Analysis of Differentially Expressed miRNAs

To investigate the possible functional role of the identified miRNAs with different expression levels in CD tumors and SCAs, we used the information on experimentally validated miRNA targets gathered in the miRtarbase release 8.0 database. High confidence known miRNA targets that were validated with luciferase reporter assay, reported in miRtarbase, were included in the analysis. The enrichment of the genes reported as miRNA targets of our 19 miRNAs of interest was determined with gene set over-representation analysis (GSOA) based on Gene Ontology (GO) Molecular Function and GO Biological Processes. The list of all the genes reported in miRTarbase as validated with reporter gene assay was used as reference. As a result, we found 30 GO Molecular Function terms and 293 GO Biological Processes terms as significantly enriched with genes that are targets of the 19 differentially expressed miRNAs. Top 10 enriched terms were related mainly to steroid hormone activity, regulation of transcription and regulation of stem cell differentiation, as shown in Figure 2. Details are presented in Appendix A. We paid special attention to the terms that refer to steroid hormone action, i.e., steroid hormone receptor activity (GO:0003707), nuclear receptor activity (GO:0004879), ligand-activated transcription factor activity (GO:0098531), as well as steroid hormone-mediated signaling pathway (GO:0043401) and hormone-mediated signaling pathway (GO:0009755). Importantly, the miRNA target genes that were overrepresented in these terms included *NR3C1* and *NR3C2* that encode for adrenal hormones glucocorticoid receptor (GR) and mineralocorticoid receptor (MR), respectively. According to the miRtarbase 9.0 database, hsa-miR-124-3p is a negative regulator of *NR3C1* gene [16] while both hsa-miR-124-3p and hsa-miR-135b-5p downregulate MR [17].

Using the PubMed search, we found additional evidence strongly supporting the role of hsa-miR-124-3p in the regulation of *NR3C1* [18,19,20,21] as well as the role of hsa-miR-135b-5p in downregulating *NR3C2* [22,23].

### 2.5. Comparison of the Expression of NR3C1 and NR3C2 in Corticotroph Adenomas Causing CD and Silent Adenomas

We determined the expression levels of *NR3C1* and *NR3C2* in corticotroph adenomas with qRT-PCR. We observed a significantly lower expression of both genes in samples from CD patients (*n* = 24) as compared to SCAs (*n* = 24); fold change (FC) 0.49 *p* = 0.0166 and FC 0.37 *p* = 0.0132, for *NR3C1* and *NR3C2*, respectively. However, the observed difference is rather slight and a notable dispersion of the results was observed (Figure 3). The differences in *NR3C1* and *NR3C2* expression correspond to the differences in hsa-miR-124-3p and hsa-miR-135b-5p levels. Patients with CD have higher levels of both miRNAs and lower levels of *NR3C1* and *NR3C2* mRNA (Figure 3). Unfortunately, we did not find a direct correlation between the expression levels of hsa-miR-124-3p and *NR3C1* or hsa-miR-135b-5p and *NR3C2*.

### 2.6. Investigtion of miRNA-Related Regulation of NR3C1 In Vitro

Transfecting the cultured cells with miRNA mimics is the commonly used approach of in vitro validation of specific miRNA–mRNA interaction. We used mice corticotroph tumor AtT-20/D16v-F2 cells for in vitro experiment and initially verified whether these cells do express *Nr3c1* and *Nr3c2* genes using deposited RNAseq data from a previous experiment on AtT-20 cells (GSE132324; Gene Expression Omnibus) and qRT-PCR. This showed that the AtT-20/D16v-F2 have relatively high expression of *Nr3c1* but do not express *Nr3c2.* Thus, we focused on the regulatory role of miR-124-3p on *Nr3c1* expression. We used miRBase [24] and Targetscan [25] to determine whether miR-124-3p is evolutionarily conserved in humans and mice and whether it targets *NR3C1* in both species. It confirmed that miR-124-3p is broadly conserved and it shares the same sequence of mature miRNA in humans and mice. Importantly, GR is among highly rated miR-124-3p predicted targets in both humans and mice and two highly conserved miR-124-3p binding motifs in 3′UTR of this gene were identified in these two species (Figure 4A).

When we transfected AtT-20/D16v-F2 cells with miR-124-3p miRNA mimic and unspecific negative control miRNA mimic, we observed a significant decrease in *Nr3c1* expression in cells treated with *miR-124-3p* miRNA mimic (Figure 4B). It was significantly lower than in cells treated with unspecific miRNA mimic. This difference was also clearly visible at the protein level. The GR level was reduced in cells treated with miR-124-3p miRNA mimic as compared to control (Figure 4B).

Two fragments of *Nr3c1* 3′UTR including each of putative miR-124-3p binding motifs were cloned in plasmid vector into 3′ region of the firefly luciferase gene. AtT-20/D16v-F2 cells were transfected with empty vector, vector with miR-124-3p binding site 1 and vector miR-124-3p binding site 2. Each of the three variants of the cells were cotransfected with miR-124-3p miRNA mimic or unspecific miRNA mimic that served as a negative control. Luminescence was developed 48 h after transfection and detected with microplate reader. As a result, we observed a significant decrease in luminescence in the cells with introduced plasmid with miR-124-3p binding site 2 treated with miR-124-3p mimic as compared to the cells transfected with the same plasmid construct but with control miRNA mimic. This observation confirms the interaction between miR-124-3p and 3′ UTR of Nr3c1 at putative binding site 2 (Figure 4C). The experiment did not confirm an interaction between miR-124-3p and 3′ UTR of Nr3c1 at binding site 1 since no significant difference of luminescence was found in cells transfected with plasmid vector harboring this binding motif treated with miR-124-3p mimic and the same cells treated with negative miRNA mimic (Figure 4C).

## 3. Discussion

Based on the clinical manifestation and biochemical tests results, pituitary corticotroph adenomas can be divided into functioning adenomas causing Cushing’s disease and SCAs. These two subtypes of tumors also differ in terms of some characteristics in MRI [2,26] and pathological features [27]. In contrast to CD-causing adenomas which are commonly small microadenomas, SCAs are diagnosed as macroadenomas due to neurological symptoms related to tumor mass. They are characterized by invasive growth, high risk of recurrence and resistance to medical therapy and are therefore referred to as “high-risk” pituitary adenomas according to current classification [3,4]. In our study, the SCAs were larger than functioning counterparts, as expected. A clear prevalence of women is observed among CD patients according to literature data [28], while it is not observed in patients suffering from SCAs. Our SCA group contained near equal representation of women and men as in previous reports [29,30]; however, some studies indicated female prevalence in SCAs [31].

Comparing functioning and silent corticotroph adenomas, we did not observe difference in patients’ age as well as differences in invasive growth status, ratio of adenomas with increased proliferation index and proportions of sparsely and densely granulated adenomas that may suggest the lack of difference in the tumors’ “aggressiveness”. Importantly, limitations for generalization of our results should be noted. The number of patients included in the analysis is relatively low and the group is not representative of the general population, especially in the case of patients suffering from Cushing’s disease. Since the main goal of our study was a molecular profiling of tumor tissue, we intentionally preselected large adenomas, which allowed us to have enough tissue for DNA/RNA isolation and successful molecular procedures.

In our investigation, we observed a negative correlation between cortisol/ACTH ratio and tumor volume in functioning corticotroph adenomas as described previously [32]. However, we did not observe any difference between micro- and macroadenomas causing CD as compared to SCAs (data not shown) as was found in previous studies [12].

The reason why some of corticotroph adenomas exhibit excessive hormone secretion and the others remain clinically silent is unclear and only few attempts have been made to determine the possible molecular mechanism underlying this difference in secretory activity. They were mainly focused on investigating the expression of the selected genes or proteins by comparing subclinical and functioning corticotroph adenomas. These studies indicated different expression levels of prohormone convertase 1/3 *POMC*, genes encoding somatostatin receptors, corticotropin releasing hormone receptor 1, vasopressin receptor (V1BR), corticosteroid 11-beta-dehydrogenase as well as NEUROD1 and TPIT [6,7,8,9,10,11,12,13]. However, whole transcriptome studies indicated that adenomas causing CD and subclinical corticotroph adenomas share a very common gene expression profile and a very low number of differentially expressed genes can be found by comparing transcriptome of silent and CD-causing ACTH tumors [14,15].

In our study, we determined the miRNA expression profile of 28 clinically functioning adenomas and 20 SCAs with next-generation sequencing of small RNA fraction. This allowed for the quantification of over 1900 miRNA annotated to current version of miRbase database and comparing their expression in two groups of tumor samples. We found a significant difference only in the expression levels of 19 miRNAs, that represent less than 1% of the miRNAs included in the analysis. This result resembles the observation from previous comparison of whole transcriptome profiles in functioning adenomas and SCAs where only 34 differentially expressed genes were found. Generally, both observations indicate a very common molecular profile of corticotroph adenomas, regardless of the functional status.

In our study, the expression levels of 13 out of 19 identified differentially expressed miRNAs were also correlated with peripheral ACTH/cortisol levels, further supporting the role of these miRNAs in secretory activity of corticotroph adenomas.

The possible role of miRNA in subclinical nature of SCAs was addressed in only one previous study by García-Martínez A et al. [33]. The authors compared the expression of 5 miRNAs in 24 functioning and 23 silent adenomas and observed a difference in hsa-miR-200a and hsa-miR-103 levels [33]. Their results were not confirmed by our investigation since these two miRNAs were not found among differentially expressed miRNAs. In our data, very a similar expression level of hsa-miR-200a was observed in clinically functioning and silent adenomas. In turn, a slightly higher expression of hsa-miR-103a-3p was observed in SCAs as previously reported, but the difference did not cross the significance threshold level. We should note that different methods were used for these two studies and technical and analytical differences could result in this discrepancy.

Since miRNAs play a role in gene regulation, their effect should be investigated in the context of the function of targeted genes. The interaction between miRNA and its target mRNA 3′UTR can be predicted with in silico tools. Unfortunately, prediction results can be very difficult to interpret since a huge number of predicted interactions can be found for some miRNAs. For example, when using the Targetescan (http://www.targetscan.org; accessed on 28 February 2022) prediction tool [25], over 4000 target genes were predicted for each hsa-miR-9-3p, hsa-miR-1909-3p, hsa-miR-22-5p and hsa-miR-181b-3p that we found as differentially expressed in CD and SCA. Therefore, to investigate a possible functional relevance of differentially expressed miRNAs we used a database of experimentally validated miRNA targets [34]. Gene set over-representation analysis of miRNA target genes indicated their enrichment in the pathways of steroid hormone nuclear receptors functioning. This result indicates that miRNAs that have different expression levels in CD and SCAs play a role in the regulation of expression of genes involved in steroid hormone signaling at hormone receptor level. It is especially interesting since this group of compounds includes adrenal hormones that play a role in the regulation of the hypothalamic–pituitary–adrenal (HPA) axis.

The particular enriched miRNA target genes included *NR3C1* and *NR3C2* that encode for corticosteroid hormone receptors (GR and MR, respectively). Both receptors are located in the cytoplasm where they bind glucocorticoids. Upon ligand binding, they are translocated to nucleus where they form dimers on DNA at glucocorticoid response elements (GREs). Glucocorticoid and mineralocorticoid receptors directly regulate the expression of target genes and/or influence the expression indirectly through the interaction with other transcription factors [35].

Glucocorticoids play a role in the basic mechanism of negative feedback of HPA axis. They act on hypothalamus, where high cortisol levels reduce secretion of corticotropin-releasing hormone (CRH), thus they directly reduce stimulation of ACTH secretion by anterior pituitary lobe. Glucocorticoids also inhibit the activity of pituitary cells indirectly. Corticotroph cells express GRs and their activation results in the reduction of *POMC* expression and secretion of ACTH [36,37]. In pituitary corticotroph adenomas, *NR3C1* point mutations and loss of heterozygosity in *NR3C1 locus* were identified [38]. These mutations seem to affect the secretory activity and result in tumor resistance to corticosteroids [39]. Reduced expression of corticosteroid receptors in corticotroph adenomas has been reported in patients with resistance to high doses of dexamethasone [40]. These data indicate a role of GR in secretory activity of clinically functioning corticotroph adenomas. The expression of corticosteroid genes was previously investigated in CD-causing tumors and SCAs and no significant differences were found. However, it is worth noting that a low number of SCA patients was included in these studies: *n* = 9 [13], *n* = 8 [11] and *n* = 2 [41].

According to previously published results, hsa-miR-124-3p is a negative regulator of *NR3C1* [16,18,19,20,21]. This was observed in acute lymphoblastic leukemia [19], adipocytes [20] and human embryonic kidney cells [21], where the reduced expression of *NR3C1* upon an increase in hsa-miR-124-3p as well as a direct interaction between this miRNA and 3′UTR of GR gene were observed. Some additional clinical observations also suggest the role of hsa-miR-124-3p in the regulation of the response to cortiosteroids in patients with acute-on-chronic liver failure [18] and lymphoblastic leukemia [19]. Hsa-miRNA-124 also mediates corticosteroid resistance in T-cells of sepsis patients through the downregulation of GR [42].

Our analysis of the expression level of *NR3C1* in corticotroph adenomas showed that tumors causing CD have lower gene expression and accordingly they exhibit higher levels of hsa-miR-124-3p. Subsequently, the role of hsa-miR-124-3p in *NR3C1* downregulation was confirmed in mice AtT-20/D16v-F2 corticotroph cells using miRNA mimics and reporter gene assay. Transfection of AtT-20/D16v-F2 cells with hsa-miR-124-3p mimics resulted in reduced *NR3C1* mRNA expression and GR protein level. We also confirmed the interaction between hsa-miR-124-3p and one of two predicted binding motifs in 3′UTR of *NR3C1* with luciferase reporter gene assay. Since sequences of hsa-miR-124-3p and target sequence in 3′UTR of NR3C1 mRNA are the same in mice and in humans, we believe that results showing the regulation of the GR-encoding gene in mice AtT-20/D16v-F2 cells are also relevant to humans. Together, the available data indicate that in pituitary corticotrophs, hsa-miR-124-3p downregulates the expression of the GR gene. Since this receptor mediates the response of pituitary cells to cortisol, the expression of hsa-miR-124-3p appears to be an important element in the regulation of secretory activity of corticotroph cells. Based on these results, we can hypothesize that in CD, a high level of hsa-miR-124-3p contributes to lowering of GR expression and in consequence it plays a role in lowering the effect of glucocorticoid feedback on the activity of corticotroph adenoma. Hsa-miR-124-3p and hsa-miR-135b-5p can downregulate the expression level of MR, as proven in model HeLa cells [17]. Expression of both miRNAs is higher in corticotroph adenomas causing CD which corresponds to the lower expression of the *NR3C2* gene in these tumors as compared to SCAs. Since the role of the MR receptor expression in pituitary cells is poorly understood, the functional implication of this observation is much less clear than in the case of GR downregulation. MR and GR have similar amino acid sequences, especially in DNA-binding domain, but they differ in affinity to corticosteroids. MR is specific for both mineralocorticoids and glucocorticoids while GR is specific predominantly for glucocorticoids. MRs have much higher affinity for glucocorticoids than GRs and are activated at basal glucocorticoid conditions, while GR occupancy is increased when glucocorticoid levels rise during the circadian peak or stress. Due to these differences, these two receptors play slightly different roles, despite the fact that they share a number of target genes [43]. MR expression is considered more tissue-specific than GR and was reported to be the most prevalent in kidney and adipose tissue but also in the hippocampus and hypothalamus [44]. However, the available databases of human expression pattern such as the Genotype-Tissue Expression project (https://gtexportal.org; accessed on 10 December 2021) or Protein atlas (https://www.proteinatlas.org; accessed on 10 December 2021) indicate that MR is widely expressed in multiple human tissues and organs including the pituitary gland. Unfortunately, a role of MR receptor in pathogenesis of pituitary tumors remains unknown.

AtT-20 cells, which are the only available cell line model of corticotroph adenoma, do not express MR receptor, thus the procedure of experimental validation of the role of miRNA in *NR3C2* silencing is not applicable. With a lack of experimental data on the exact role of MR, we can only hypothesize that miRNA-mediated silencing of *NR3C2* may have the similar effect on HPA axis feedback as silencing of *NR3C1*. It may enhance ACTH secretion by reducing the direct inhibitory effect of glucocorticoids on neoplastic pituitary corticotrophs.

The difference in expression of hsa-miR-124-3p and hsa-miR-135b-5p between subclinical and CD-causing adenomas is not big, thus we suppose that high expression of these miRNAs is not the only cause of difference in ACTH secretion. Presumably this is one of the mechanisms in the regulation of corticotrophs’ secretory activity. The model of miRNA-based corticosteroid receptor regulation does not undermine the role of previously described differences in the expression of convertase 1/3, *POMC*, somatostatin receptors or corticotropin releasing hormone receptor 1 or genes involved in differentiation of pituitary cells [6,7,8,9,10,11,12,13]. When considering the complex nature of the regulation of ACTH secretion, it can be assumed that multiple mechanisms may be involved in the silent character of subclinical adenomas. The low number of identified differentially expressed miRNAs or genes in silent and clinically functioning adenomas probably results from the intertumoral molecular heterogeneity of SCAs. This is also in line with clinical evidence indicating that some silent corticotroph adenomas can transform into clinically functioning ones while the others remain silent [1].

The misregulation of GR expression or *NR3C1* mutation may have important therapeutical implications in CD patients. Non-selective GR antagonist Mifepristone was officially approved for treatment in patients with Cushing’s syndrome [45] while another new GR inhibitor, Relacorilant (CORT125134), is under clinical investigation for its use in this group of patients [46]. The further studies will be required to assess the role of GR abnormalities in response to GR-targeting treatment in CD.

In our study, we focused mainly on the role of hsa-miR-124-3p and hsa-miR-135b-5p in the regulation of corticosteroid receptors, but the role of other differentially expressed miRNAs can also be elucidated, based on the function of putative target genes. In the pathways enrichment analysis of the putative targets, molecular functions related to transcriptional regulation were found among the top processes. Interestingly, five miRNAs, i.e., hsa-miR-132-5p, hsa-miR-135b-5p, hsa-miR-27a-3p, hsa-miR-9-3p and hsa-miR-9-5p, were previously reported to downregulate the expression of *FOXO1* transcription factor [47,48,49,50,51]. FOXO1 plays an important role in the differentiation of pituitary cells [52] and secretion of gonadotropic hormones [53,54] and prolactin [55]. The role of *FOXO1* in pituitary corticotroph cells was not investigated but it was shown to regulate *POMC* expression in POMC hypothalamic neurons [56]. In POMC, neurons of arcuate nucleus FOXO1 directly suppresses *POMC* expression. A similar mechanism was also observed in prolactin pituitary adenomas where FOXO1 suppresses the promoter of *PRL* gene [55]. It is possible that high expression of hsa-miR-132-5p, hsa-miR-135b-5p, hsa-miR-27a-3p, hsa-miR-9-3p and hsa-miR-9-5p in pituitary corticotroph adenomas reduces the level of FOXO1 and eventually contributes to the upregulation of *POMC* expression. In our data from corticotroph adenomas, we observed the correlation between levels of hsa-miR-9-3p/hsa-miR-9-5 and *POMC* expression, which also supports this concept, but the exact role of miRNAs in possible FOXO1-related regulation of secretory activity of corticotroph cells requires further functional investigation.

## 4. Materials and Methods

### 4.1. Patients and Tissue Samples

Pituitary tumor samples from 48 patients were collected during transsphenoidal surgery. Formalin-fixed and paraffin-embedded (FFPE) tissue samples, including 28 samples from patients with Cushing’s disease and 20 samples of SCA were used for the study. Diagnosis of hypercortisolism was based on standard hormonal criteria: increased UFC in three 24 h urine collections, disturbances of cortisol circadian rhythm, increased serum cortisol levels accompanied by increased or not suppressed plasma ACTH levels at 8.00 and a lack of suppression of serum cortisol levels to <1.8 µg/dL during an overnight dexamethasone suppression test (1 mg at midnight). The pituitary etiology of Cushing’s disease was confirmed based on the serum cortisol levels or UFC suppression < 50% with a high-dose dexamethasone suppression test (2 mg q.i.d. for 48 h) or a positive result of a corticotrophin-releasing hormone stimulation test (100 mg i.v.) and positive pituitary magnetic resonance imaging.

ACTH levels were assessed using IRMA (ELSA-ACTH, CIS Bio International, Gif-sur-Yvette Cedex, France). The analytical sensitivity was 2 pg/mL (reference range: 10–60 pg/mL). Serum cortisol concentrations were determined by the Elecsys 2010 electrochemiluminescence immunoassay (Roche Diagnostics, Mannheim, Germany). Sensitivity of the assay was 0.02 μg/dL (reference range: 6.2–19.4 μg/dL). UFC was determined after extraction (liquid/liquid with dichloromethane) by electrochemiluminescence immunoassay (Elecsys 2010, Roche Diagnostics)—reference range: 4.3–176 μg/24 h.

All the tumors underwent detailed histopathological diagnosis including immunohistochemical staining with antibodies against particular pituitary hormones (ACTH, GH, TSH, FSH, LH, α-subunit) and Ki67 as well as ultrastructural analysis with electron microscopy.

The SCAs were characterized by the following clinicopathological criteria: positive immunohistochemical staining for ACTH, lack of signs and symptoms of hypercortisolism (Cushing’s syndrome), negative hormonal evaluation and non-compliance with diagnostic criteria of the CD.

Macroadenoma was defined as an adenoma with at least one diameter exceeding 10 mm, and the tumor volume was assessed with the diChiro Nelson formula (height × length × width × π/6). Invasive growth of the tumors was evaluated using Knosp grading [57]. Adenomas with Knosp grades 0, 1 and 2 were considered non-invasive, while those with Knosp 3 and 4 were considered invasive.

Forty-three patients had a clear history of not using any drugs that control the overproduction of the cortisol or ACTH (ketoconazole, mitotane, metyrapone, osilodrostat, mifepristone, pasireotide) before surgical treatment. The information on preoperative pharmacological treatment was not available for 5 patients.

Tumor tissue content of each FFPE sample ranged between 80 and 100% (median 99%), as assessed with histopathological examination. Patients’ characteristics are presented in Table 1 and details on each patient’s data are available in Appendix A.

The study was approved by the local Ethics Committee of Maria Sklodowska-Curie National Research Institute of Oncology in Warsaw, Poland. Each patient provided informed consent for the use of tissue samples for scientific purposes.

Total RNA from FFPE samples was purified with RecoverAll™ Total Nucleic Acid Isolation Kit for FFPE tissue (Thermo Fisher Scientific, Waltham, MA, USA) and measured using NanoDrop 2000 (Thermo Fisher Scientific). RNA was stored at −70 °C.

### 4.2. Micro RNA Expression Profiling

For comparing the miRNA expression profiles in CD-causing and clinically silent adenomas, NGS data from our previous investigation of miRNA expression in corticotroph adenomas were used. The dataset is available at Gene Expression Omnibus, accession no GSE166279. Sequencing of small RNA fraction was performed in 48 tumor samples (28 CD patients and 20 SCA patients) with ion semiconductor sequencing technology, as described previously [58]. Briefly, Ion Total RNA-Seq Kit v2 (Thermo Fisher Scientific) was used for sequencing library construction, Ion Xpress™ RNA-Seq Barcode Kit was used for hybridization and ligation of RNA adapters. RNA reverse transcription and subsequent cDNA purification and library size selection were performed using Nucleic Acid Binding Beads and verified using Bioanalyzer 2100 with High Sensitivity DNA Kit (Agilent, Santa Clara, CA, USA). Ion Chef instrument, with Ion PI™ Hi-Q™ Chef Kit (Thermo Fisher Scientific) and Ion Proton sequencer (Thermo Fisher Scientific) were used for library preparation and sequencing, respectively.

BamToFastq package was applied for converting unmapped bam files into fastq files. miRDeep2 was applied for read mapping to known human miRNAs (according to miRBase release 22) and reads quantification. Data normalization and differential expression analysis were performed using DESeq2. Filtration for low-expression miRNAs was applied as described previously. FC of expression calculated as the ratio of the normalized read-count value in CD-causing and silent adenomas was used as a measure of expression difference. Adjusted *p*-value < 0.05 was used as significance threshold. MiRtarbase release 9.0 database [34] was used to identify known miRNA target genes. PANTHER (http://pantherdb.org; accessed on 10 December 2021) [59] was used for gene set over-representation analysis.

### 4.3. qRT-PCR gene Expression Analysis

One microgram of RNA was subjected to reverse transcription with Transcriptor First Strand cDNA Synthesis Kit (Roche Diagnostics). qRT-PCR reaction was carried out in 384-well format using 7900HT Fast Real-Time PCR System (Applied Biosystems, Foster City, CA, USA) and Power SYBR Green PCR Master Mix (Thermo Fisher Scientific) in a volume of 5 μL, containing 2.25 pmol of each primer. The samples were amplified in triplicates. *GAPDH* was used as reference gene. Delta Ct method was used to calculate the relative expression level. PCR primers’ sequences are presented in Appendix A.

### 4.4. Cell Line Culture and miRNA Mimic Transfection

AtT-20/D16v-F2 cells were purchased from ATCC collection and cultured in DMEM medium supplemented with 10% FBS, as recommended. MiRCURY LNA miRNA Mimics including hsa-miR-124-3p mimic (YM00471256, Qiagen, Hilden, Germany) and negative control mimic (YM00479902-ADB, Qiagen) were used. AtT-20/D16v-F2 cells were seeded at 5 × 10^4^ per well of a 24-well plate in culture medium and transfected with 50 nM miRNA with 1% (*v*/*v*) HiPerFect Transfection Reagent (Qiagen), according to the manufacturer’s instructions. The next day, the culture medium was changed. In total, 48 h after transfection the cells were harvested and subjected to isolation of total RNA with RNeasy Mini Kit (Qiagen). The expression of the putative hsa-miR-124-3p target gene was determined with qRT-PCR.

### 4.5. Luciferase Reporter Gene Assay

Hsa-miR-124-3p target sites in 3′UTR of NR3C1 were determined with Targetscan [25]. Each of two predicted hsa-miR-124-3p target sites were cloned into pmirGLO Dual-Luciferase miRNA Target Expression Vector (Promega, Madison, WI, USA). AtT-20/D16v-F2 cells (2 × 10^4^/well) were seeded onto a 96-well plate in 100 µL culture medium. The next day, the cells were transfected with 100 ng of each plasmid vector, independently using 0.25% (*v*/*v*) lipofectamine 3000 (Invitrogen, Carlsbad, CA, USA) in 10 µL of DMEM. The cells were subsequently transfected with either hsa-miR-124-3p mimic (YM00471256, Qiagen) or negative control mimic (YM00479902-ADB, Qiagen) in a final concentration of 50 nM using HiPerfectReagent (Qiagen). Culture medium was changed on the next day. Luciferase activity was measured with One-Glo Luciferase Assay System (Promega) 48 h after transfection.

### 4.6. Western Blotting

Cells were lysed in ice cold RIPA buffer, incubated for 30 min in 4 °C and centrifuged at 12,500× *g* rpm for 20 min at 4 °C. Samples were resolved using SDS-PAGE and electrotransferred to polyvinylidene fluoride membranes (PVDF) (Thermo Fisher). GR protein was detected with monoclonal anti-Glucocorticoid Receptor antibody (ab183127, Abcam, Cambridge, UK), and secondary anti-rabbit antibody conjugated to HRP (#7074, Cell Signaling, Beverly, MA, USA). Glyceraldehyde-3-Phosphate Dehydrogenase (#MAB374, Millipore, Bedford, MA, USA) detected with mouse HRP-conjugated antibody (#7076 Cell Signaling) served as control. Visualization was performed with SuperSignal West Pico Chemiluminescent Substrate (Thermo Fisher Scientific) and CCD digital imaging system Alliance Mini HD4 (UVItec Limited, Cambridge, UK).

### 4.7. Statistical Analysis

A two-sided Mann–Whitney U-test was used for analysis of continuous variables. The Spearman correlation method was used for correlation analysis. Significance threshold of α = 0.05 was adopted. Data were analyzed using GraphPad Prism 6.07 (GraphPad Software, La Jolla, CA, USA). Hierarchical clustering analysis was carried out with Cluster 3.0, and the results were visualized using TreeView 1.6 software (Stanford University School of Medicine, Stanford, CA, USA).

## Figures and Tables

**Figure 1 ijms-23-02867-f001:**
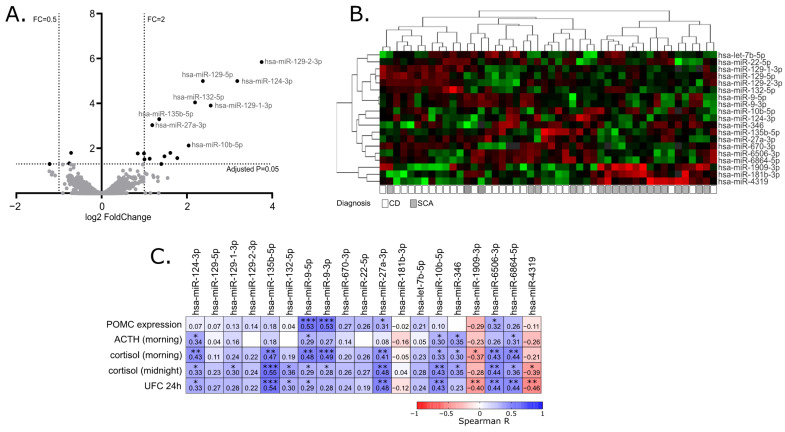
MiRNA expression profiling in corticotroph adenomas. (**A**). Difference in miRNA expression between functioning and silent corticotroph adenomas. Volcano plot showing differentially expressed miRNAs. Significance and fold change thresholds are marked with dashed lines. (**B**). Heat map representing the expression of differentially expressed miRNAs and clustering the samples of adenomas causing Cushing’s disease (CD) and silent corticotroph adenomas (SCA). (**C**). The correlation between the expression levels of differentially expressed miRNAs and *POMC* expression or hormonal laboratory measurements in patients: morning plasma ACTH level, morning and midnight plasma cortisol levels and 24 h urinary free cortisol; * indicate *p*-value < 0.05; ** indicate *p*-value < 0.01; *** indicate *p*-value < 0.001

**Figure 2 ijms-23-02867-f002:**
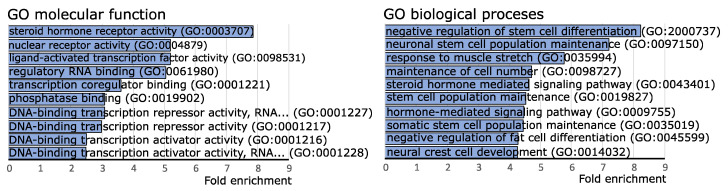
Gene set over-representation analysis of putative target genes of miRNAs differentially expressed in clinically functioning and silent corticotroph adenomas.

**Figure 3 ijms-23-02867-f003:**
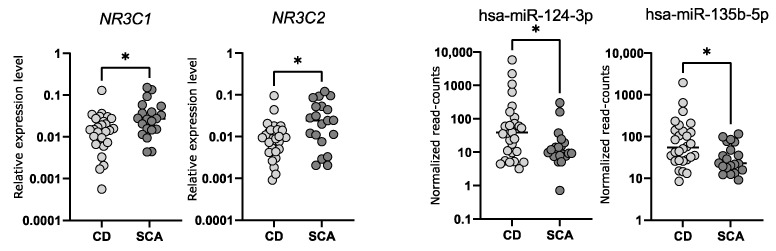
The expression levels of *NR3C1* and *NR3C2* measured with qRT-PCR as well as hsa-miR-124-3p and hsa-miR-135b-5p measured with small RNA sequencing in tumor samples from CD patients and silent corticotroph adenomas; * indicate *p*-value < 0.05

**Figure 4 ijms-23-02867-f004:**
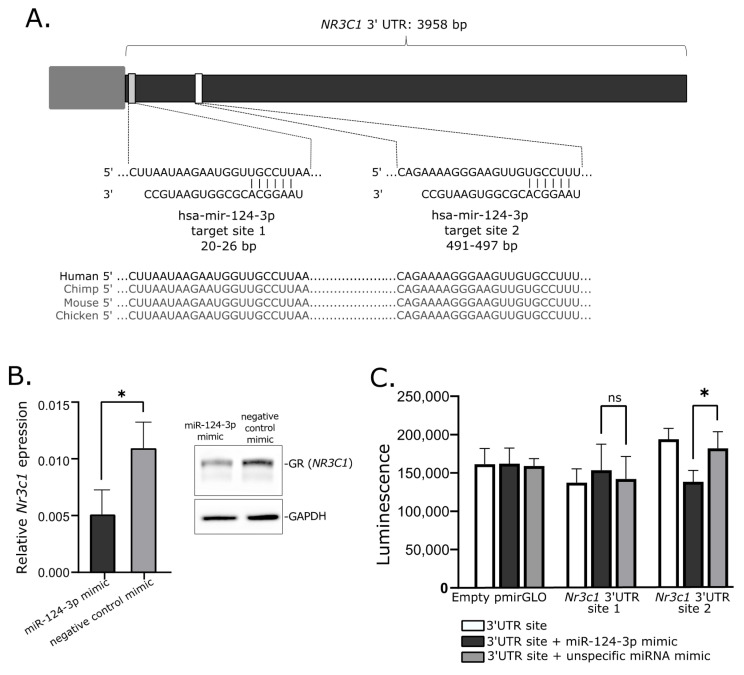
Role of mir-124-3p in regulation of glucocorticoid receptor gene. (**A**). Putative hsa-mir-124-3p target sites in 3′UTR of *NR3C1*. (**B**). Reduced expression of *Nr3c1* gene expression and glucocorticoid receptor (GR) protein level in AtT-20/D16v-F2 cells treated with hsa-miR-124-3p mimics. (**C**). Results of luciferase reporter gene assay, showing the interaction between *Nr3c1* 3′UTR site 2 and mir-124-3p; * indicate *p*-value < 0.05; ns—not significant.

**Table 1 ijms-23-02867-t001:** Summary of clinical features of patients with Cushing’s disease and silent corticotroph adenomas.

Clinical Feature	Cushing’s Disease	Silent Corticotroph Adenomas	
Number of patients	*n* = 28	*n* = 20	
Sex (ratio females/males)	25/3	11/9	*p* = 0.0156
Age at surgery (years; median (range))	42 (23–76)	54 (23–77)	ns
Cortisol 08:00 h (µg/dL; median (range))	26.3 (11.9–49.7)	18.15 (6.8–50)	*p* = 0.0002
Cortisol 24:00 h (µg/dL; median (range))	19.8 (11.6–36.5)	0.9 (0.3–4.9)	*p* < 0.0001
ACTH 08:00 h (pg/dL; median (range))	55 (36.9–129)	47.7 (14.7–74.9)	ns
UFC (μg/24 h; median (range))	493.5 (215–810)	96.08 (13.7–130)	*p* < 0.0001
Tumor volume (mm^3^; median (range))	713 (32–6750)	3420 (900–11088)	*p* = 0.0008
Invasive tumor growth (Knosp grade 0, 1, 2/3, 4)	21/7	16/4	ns
Histology (sparsely/densely granulated)	9/19	12/8	ns

**Table 2 ijms-23-02867-t002:** The list of miRNAs differentially expressed in corticotroph pituitary adenomas causing CD and silent corticotroph adenomas.

	MiRNA ID	Fold Change	Adjusted *p*-Value
**1.**	hsa-miR-129-2-3p	13.46395457	1.43 × 10^−6^
**2.**	hsa-miR-129-5p	10.19063213	1.00 × 10^−5^
**3.**	hsa-miR-124-3p	9.070928502	1.00 × 10^−5^
**4.**	hsa-miR-132-5p	4.57209911	9.08 × 10^−5^
**5.**	hsa-miR-129-1-3p	5.895678939	0.000124961
**6.**	hsa-miR-135b-5p	2.551718618	0.000501695
**7.**	hsa-miR-27a-3p	2.276970705	0.000935298
**8.**	hsa-miR-10b-5p	4.115713962	0.007517745
**9.**	hsa-miR-9-3p	3.053715906	0.01591924
**10.**	hsa-miR-1909-3p	0.608506895	0.01591924
**11.**	hsa-miR-6506-3p	1.99399423	0.017071361
**12.**	hsa-miR-6864-5p	1.799457096	0.017071361
**13.**	hsa-let-7b-5p	2.780063988	0.022598034
**14.**	hsa-miR-670-3p	3.421342816	0.027286789
**15.**	hsa-miR-22-5p	2.187802851	0.028881635
**16.**	hsa-miR-346	2.006887582	0.030967044
**17.**	hsa-miR-4319	0.591700763	0.04665918
**18.**	hsa-miR-181b-3p	0.42948911	0.049613422
**19.**	hsa-miR-9-5p	2.641857365	0.049732716

## Data Availability

Data from next-generation sequencing of small RNA fraction of 48 corticotroph adenoma samples are available at Gene Expression Omnibus, accession no GSE166279.

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
