# Peer review of "Difference in miRNA Expression in Functioning and Silent Corticotroph Pituitary Adenomas Indicates the Role of miRNA in the Regulation of Corticosteroid Receptors"

_ijms, 2022, doi:10.3390/ijms23052867_

Round 1
Reviewer 1 Report
This manuscript investigates the miRNA expression in functioning and silent corticotroph pituitary adenomas, based on samples from 48 patients operated by transsphenoidal surgery.
Although the results provide some noticeable information, the small number of cases represents the weak point of this study. In this way, the strength of their conclusions is not sufficient to support the importance of miRNA expression in order to elucidate the underlying mechanisms.
In addition, there are some other issues to consider.
According to the WHO classification of 2017, silent corticotroph adenomas belong to the new category of “High-risk” pituitary adenomas. Also, there are recent reviews on this category presenting the histological and clinical characteristics of these adenoma types and discussing the challenges in diagnosis, prognosis and treatment. The authors should take into consideration this information.
Given that functioning and silent corticotroph adenomas are identical by histology and hormone immunohistochemistry, Galectin-3 represents the only distinguishing immunohistochemical marker. The authors should comment on that and include the relevant references.
In a paper, already quoted in this study, Raverot et al 2010, have shown that patients with silent corticotroph adenomas show normal concentration of cortisol, but elevated ACTH plasma levels, similar to functioning microadenoma counterparts. Also, the cortisol/ACTH ratio is similar to functioning corticotroph microadenomas, but lower than in functioning macroadenomas. This may suggest a slight different regulation of glucocorticoid receptors.
In the discussion, it is not clear whether the authors agree with the validity of these findings.
Finally, the term “PitNETs” is a proposal by a small group in an attempt to replace the original term “pituitary adenoma”, which by definition is of neuroendocrine origin. However, this attempt was not acceptable, at least for Neurosurgeons and many Endocrinologists. Therefore, my suggestion is to replace PitNETs through the text and revert the original term “pituitary adenoma” instead. In that case, the specified term “neuroendocrine” is not required. The “PitNETs”, should be mentioned only as a proposed alternative term.
The reference list is incomplete. The authors need to amend it.
Reviewer 2 Report
The authors designed a study to find markers that differentiated corticotroph with ACTH secretion with clinical Cushing Disease from silent corticotroph tumors. For this, they used data from small RNA sequencing in tumors. The authors found similar molecular profile in both tumors types, regardless the functional status.
Limitations:
- Corticotroph linage origin it is not assed in the text. How the authors can ensure the corticotroph origin? The authors only mention how they check ACTH secretion (confirming biochemically Cushing disease) but they never explain the Pathology characteristics Histology (sparsely/densely granulated) is not a marker of ACTH linage
- Glucocorticoids influence cell proliferation, survival, function and probably also regulates microRNA expression, therefore the change in microRNA could be a result of hypercortisolism and not due to tumor expression
- In the case of clinical Cushing there is a lack information of previous drug intake ( pre-surgery cortisol levels normalization, could explain the lack of differences between these very different tumors
- The authors mention a previous study (reference 22) with different results, but they did not argue, where this differences could lie?
- Because of the lack of differences, the authors attempt to focus on glucocorticoids receptors, but all the theory of downregulations of receptors, in clinical Cushing it is explained by glucocorticoid feedback rather than microRNA differential expression.
Strengths:
- Clinical interest.
- The sample number is fine.
- The redaction is clear
I think that, before proceeding with further review the authors should better describe the Pathology sample and argue all the limitations points.
Reviewer 3 Report
The manuscript entitled "Difference in miRNA expression in functioning and silent corticotroph neuroendocrine pituitary tumors indicate the role of miRNA in regulation of corticosteroid receptors" is written and designed in latest scientific credits and can be published. However, conclusion needs further refining with proper focus on the future aspects.
Round 2
Reviewer 1 Report
The authors of this manuscript have made most of the suggested changes.
However, there are still some other issues to consider.
The term “tumor” is for macroscopic description, not for histological classification. Therefore, the authors should replace it throughout the manuscript by the term “adenoma”; e.g. :“corticotroph tumors” should be “corticotroph adenomas”.
The authors need to update the reference list:
In addition to the quoted new reference [no 3], the reference of the book “WHO classification of Tumors of Endocrine Organs”, initially introducing the new term “High–risk” adenomas, should also be included (Osamura RY et al., vol 10, 4th edn. IARC, Lyon, p 13, 2017).
The added reference for Galectin-3 is out of date. It should be replaced by a more recent publication reporting that Galectin-3 is the only immunohistochemical marker distinguishing silent from functioning corticotroph adenomas (Hormones 6:227-32,2007 by Thodou E et al).
Author Response
Reviewer’s comment 1.The term “tumor” is for macroscopic description, not for histological classification. Therefore, the authors should replace it throughout the manuscript by the term “adenoma”; e.g. :“corticotroph tumors” should be “corticotroph adenomas”.
Reply: We changed “corticotroph tumors” into “corticotroph adenomas” through the manuscript, as requested
Reviewer’s comment 2. In addition to the quoted new reference [no 3], the reference of the book “WHO classification of Tumors of Endocrine Organs”, initially introducing the new term “High–risk” adenomas, should also be included (Osamura RY et al., vol 10, 4th edn. IARC, Lyon, p 13, 2017).
Reply. We introduced the reference to WHO classification book, as suggested. (reference 4
Reviewer’s comment 3. The added reference for Galectin-3 is out of date. It should be replaced by a more recent publication reporting that Galectin-3 is the only immunohistochemical marker distinguishing silent from functioning corticotroph adenomas (Hormones 6:227-32,2007 by Thodou E et al).
Reply. We replaced the referene “Jin, L.; Riss, D.; Ruebel, K.; Kajita, S.; Scheithauer, B.W.; Horvath, E.; Kovacs, K.; Lloyd, R. v Galectin-3 Expression in Functioning and Silent ACTH-Producing Adenomas; 2005; Vol. 16” with “Thodou, E.; Argyrakos, T.; Kontogeorgos, G. Galectin-3 as a Marker Distinguishing Functioning from Silent Corticotroph Adenomas. Hormones (Athens, Greece) 2007, 6, 227–232.”
Reviewer 2 Report
The authors have made a reasonable effort at revising their manuscript. I have no additional comments.
Author Response
-